# An Analysis of Burnout, Coping, and Pulse Wave Velocity in Relation to the Workplace of Healthcare Workers for the Sustainability of the Medical Career

Ioana Marin [1], Corneluta Fira-Mladinescu [2,*], Catalin Nicolae Marin [3], Victoria Stan [4] and Sorin Ursoniu [5]

1   Center of Studies in Preventive Medicine, Discipline of Occupational Medicine, Department of Internal Medicine V, Victor Babes University of Medicine and Pharmacy, Eftimie Murgu Square, No. 2, 300041 Timisoara, Romania; ioana.marin@umft.ro
2   Center of Studies in Preventive Medicine, Discipline of Hygiene, Department of Microbiology, Victor Babes University of Medicine and Pharmacy, Eftimie Murgu Square, No. 2, 300041 Timisoara, Romania
3   Department of Physics, Faculty of Physics, West University of Timisoara, V. Parvan Ave., No. 4, 300223 Timisoara, Romania; catalin.marin@e-uvt.ro
4   Faculty of Psychology and Educational Sciences, Spiru Haret University, Fabricii Street No. 46G, 060821 Bucharest, Romania; sp.stan.victoria@spiruharet.ro
5   Center for Translational Research and Systems Medicine, Discipline of Public Health, Department of Functional Sciences, Victor Babes University of Medicine and Pharmacy, Eftimie Murgu Square, No. 2, 300041 Timisoara, Romania; sursoniu@umft.ro
*   Correspondence: fira-mladinescu.corneluta@umft.ro

**Abstract:** The assessment of the health status of healthcare workers (HWs) is related to the growing interest in the sustainability of the medical profession. This study explores for the first time the level of burnout, coping strategies, and pulse wave velocity (PWV), the correlations between them, and possible connections with the workplace. It included 154 HWs, aged 25–64 years, 83.76% women, divided into five groups (oncology, cardiology, intensive care, occupational health, and residents). The Burnout Assessment Tool (BAT-23) questionnaire for the burnout level and the Carver Coping Orientation to Problems Experienced (COPE) questionnaire for the coping strategies were used. PWV was measured with a TensioMed arteriograph. Regarding BAT subscales, all studied groups are at burnout for *exhaustion (E)*, prone to burnout for *mental distancing (MD)*, *emotional impairment (EI)*, and no burnout for *cognitive impairment (CI)*. On the four subscales of the BAT, the MANOVA shows a medium-sized effect of the workplace on MD and on CI. All subjects use positive coping strategies. Problem-solving mechanisms are often used, followed by emotional support-based and social support-based coping. PWV correlates with age and BMI and has a small positive correlation with MD, CI, and EI. High burnout levels are related to the healthcare profession, not to the workplace. Our research shows the practical need to supervise the mental and physical health of HWs in order to preserve their health through medical and organizational methods.

**Keywords:** burnout; burnout assessment tool (BAT); pulse wave velocity (PWV); coping; healthcare workers (HWs); workplace; sustainability

## 1. Introduction

Since 2019, in the 11th revision of the International Classification of Diseases (ICD-11), burnout has been included as an occupational phenomenon [1]. Healthcare workers, including doctors, nurses, and other medical staff members, are at a high risk of experiencing burnout due to the demands of their jobs. Burnout is defined as a state of emotional exhaustion, depersonalization, and reduced personal achievement that results from chronic stress related to work, combined with emotional and cognitive impairments [2]. Apart from the fact that it can induce depression or other associated diseases [3,4], burnout can lead to reduced empathy, increased cynicism, and decreased engagement, all of which can

negatively impact patient care. Factors that contribute to burnout include heavy workloads, long hours, inadequate resources, high patient demands, and limited support from management and colleagues [5]. Differences in burnout were reported by specialty; for example, in the USA, in a study published in 2012 [5], emergency medicine and general internal medicine had the highest burnout rates (around 60% of respondents), while dermatology, preventive medicine, occupational health, and environmental medicine had the lowest rates (around 30% of respondents) [5]. In Ref. [6], the authors reported the precursors of the burnout syndrome in oncology nurses from a Swiss oncology institute. Burnout levels were assessed using the Burnout Potential Inventory (BPI) questionnaire, and it was found that inpatient nurses showed a higher risk of burnout than ambulatory nurses. In a study conducted during the COVID-19 pandemic in Spain among direct care workers in nursing homes using the Maslach Burnout Inventory (MBI), the results show that 6.4% of the respondents were burned out, 53.8% of the participants were emotionally exhausted, and 35.1% were found to suffer from depersonalization [7]. Another study that was also carried out with the Maslach Burnout Inventory (MBI) on nurses in Poland during the COVID-19 pandemic shows that the professional burnout of nurses depends on the hospital environment and age [8]. In surgical departments, nurses were younger, less exhausted, and handled stress better than nurses in non-invasive and intensive care units [8].

Coping refers to the psychological and behavioral efforts made by individuals to manage stress, adversity, and challenging situations effectively [9]. Life is full of uncertainties, and everyone faces difficulties at some point, be it personal, professional, or health-related issues. Coping mechanisms are essential to helping people deal with these difficulties and maintain emotional well-being. The medical profession is one of the most demanding and emotionally taxing fields, requiring healthcare workers to provide care and support to patients while facing numerous challenges. From long work hours to high-stress environments, medical workers must find effective coping strategies to maintain their well-being and continue to provide quality care [10]. For example, Gniewek et al. [8], using a COPE questionnaire with 14 coping strategies, showed that nurses who used better coping strategies, endured stress more easily and had a lower level of burnout than the others.

Coping not only impacts an individual's mental health but also has a direct influence on their physical health. Research has shown that people who use adaptive coping mechanisms tend to have better cardiovascular health, lower rates of chronic diseases, and a higher quality of life [11].

Cardiovascular diseases (CVDs) are the leading cause of death globally, according to the World Human Organization (WHO), and an estimated 17.9 million people died from CVDs in 2019, representing 32% of all global deaths [12]. Pulse wave velocity (PWV) is a measure of arterial stiffness [13]; it is correlated with CVD and is considered a predictor factor of major adverse cardiovascular events (MACEs) [14]. One of the significant advantages of PWV is that it can be measured noninvasively, making it a more practical and widely applicable method for assessing cardiovascular health.

The purpose of this prospective observational study is to assess the level of burnout syndrome, coping strategies, and their correlation with PWV among healthcare workers. This study impacts the quality assurance of the medical act and the sustainability of the medical career in different specialties at university clinics in Timisoara, Romania. Research was carried out towards the end of the COVID-19 pandemic, after restrictions specific to the pandemic were removed, when the severity of the clinical manifestations of SARS-CoV-2 infection was lower, from the end of April to October 2022. In this way, we assume that the stress, burnout, and coping tests were not influenced by mental tension or overwork at work, specific to the pandemic period.

## 2. Materials and Methods

Participants willingly enrolled, based on informed consent, after the study's purpose was presented and explained to them in person. The study was carried out according to the Declaration of Helsinki and was approved by the Ethics Committee of the University

of Medicine and Pharmacy, Timisoara, No. 59/22.12.2021. After consenting, participants responded in pen and paper to sociodemographic questions (sex, age, height, weight, and BMI) to questions regarding occupation (workplace, specialty, and work experience), to the Burnout Assessment Tool-long version (BAT-23) and Carver Coping Orientation to Problems Experienced Inventory Questionnaire. To measure PWV, a tensiomed arteriograph was used, in compliance with the following recommendations [15]: subjects were instructed in advance not to consume alcohol the night before, not to perform physical exercises, and not to smoke or drink coffee at least 3 h before the measurements. Each subject rested for 10 min in a quiet room, then the circumference of the dominant arm and the length of the distance between the upper part, the curvature of the sternum, and the upper edge of the pubis were measured, and the values were introduced in the arteriograph software v2.00.01. PWV was measured in the supine position, with the cuff on the dominant arm.

The total number of participants was 154 medical workers from university clinics in Timisoara, Romania (83.76% women). Five study groups were formed: four according to the medical specialty and a group of resident medical doctors from the same clinics. Each study group (from the oncology, cardiology, intensive care unit, and occupational health groups) consisted of physicians, nurses, and other medical personnel. The characteristics of the study groups are presented in Table 1.

**Table 1.** Characteristics of the subjects in the study groups.

| Parameter | | Oncology | Cardiology | Intensive Care Unit | Occupational Health | Residents |
|---|---|---|---|---|---|---|
| Number of subjects | | 33 | 25 | 33 | 23 | 40 |
| Percent of males | | 6.06% | 16% | 24.24% | 0% | 27.5% |
| Occupation | doctors | 9 (27.27%) | 8 (32%) | 6 (18.18%) | 7 (30.43%) | 40 (100%) |
| | nurses | 22 (66.66%) | 17 (68%) | 24 (72.72%) | 15 (65.21%) | - |
| | other medical staff | 2 (6.07%) | 0% | 3 (9.1%) | 1 (4.34%) | - |
| Age (Mean ± SD) (years) | | 38.27 ± 9.99 | 42.88 ± 10.67 | 39.36 ± 11.56 | 50.30 ± 6.51 | 27.87 ± 2.20 |
| Work experience (Mean ± SD) (years) | | 13.39 ± 9.99 | 19.32 ± 11.57 | 14.88 ± 10.83 | 27.39 ± 9.14 | 3.05 ± 2.59 |
| Body Height (Mean ± SD) (m) | | 1.65 ± 0.06 | 1.67 ± 0.11 | 1.67 ± 0.08 | 1.64 ± 0.06 | 1.70 ± 0.09 |
| Body weight (Mean ± SD) (kg) | | 67.30 ± 10.94 | 73.76 ± 17.53 | 68.18 ± 12.33 | 70.74 ± 9.83 | 66.05 ± 14.42 |
| BMI * (Mean ± SD) (kg/m$^2$) | | 24.85 ± 3.71 | 26.34 ± 4.79 | 24.39 ± 3.94 | 26.41 ± 3.39 | 22.55 ± 3.61 |

* BMI—body mass index.

In order to assess the level of burnout among the studied subjects, all participants filled out the Romanian-validated 23-item version of the Burnout Assessment Tool (BAT-23) [16]. The new questionnaire conceives burnout as a syndrome; it consists of a total core score with interrelated four components (exhaustion, mental distancing, cognitive impairment, and emotional impairment) that can also be analyzed according to the cutoffs recently established by prof. Schaufeli [17], which are used as working hypotheses in this study.

The four subscales are evaluated through four groups of questions: (a) emotional exhaustion (eight items, e.g., "*At work, I feel mentally exhausted*"), (b) cognitive impairment (five items, e.g., "*At work it happens that I overreact unwillingly*"), (c) emotional impairment (five items, e.g., "*I feel a strong aversion towards my job*"), and (d) mental distancing (five items, e.g., "*When I work, I hardly concentrate*"). Items are scored on a five-point Likert scale ranging from 1 (never) to 5 (always) [16,17]. The general cutoff values available for the European

population that allow us to distinguish between the three possibilities (no burnout, prone to burnout, and at burnout) were used for the studied groups [18].

In order to evaluate coping methods in the studied groups, the Carver Coping Orientation to Problems Experienced (COPE) Inventory Questionnaire [19] was used. This COPE Questionnaire was developed by Carver, Scheier, and Weintraub [19] and was translated, adapted, and validated in Romania. The Romanian version of COPE demonstrated high internal consistency, with a Cronbach's alpha coefficient of 0.74 [20]. The COPE questionnaire covers 15 forms of coping that include 60 statements, with each form of coping being evaluated through 4 items.

Each item is rated on a four-point Likert scale, with the following meanings: 1—"*I usually do not do this*"; 2—"*I rarely do this*"; 3—"*I sometimes do this*"; 4—"*I often do this*". Participants had to indicate the degree to which they agreed with each item according to that Likert-type scale. The score of each dimension is presented as the sum of its constituent elements divided by the number of elements. The highest scaled score shows which method of coping is frequently used.

The fifteen scales corresponding to coping strategies related to responses in stressful situations are divided into four categories [20]: (a) coping focalized on emotion (positive reinterpretation and growth, acceptance and restraint, religious approach); (b) coping focused on problem (active approach, avoiding competing activities and planning); (c) coping focused on social support (search for social instrumental and/or social emotional support and emotional discharge); and (d) avoidant coping (denial, mental passivity, and behavioral passivity). Other coping scales evaluate substance consumption (alcohol/drug consumption) and humor (the traumatic situations are presented in an ironic manner).

Statistical processing of the results was performed using SPSS software, 18.0 version. Means, standard deviations, multivariate analysis of variance (MANOVA), and Pearson's correlation coefficients are presented. The statistical significance was established at $p < 0.05$.

## 3. Results

### 3.1. Results of the BAT-23 Test for the Study Groups

The results of the Burnout Assessment Tool test (BAT-23) for the study groups are presented in Tables 2 and 3. Table 2 includes the mean values and standard deviations (SD) for the total core score and for the BAT subscales (exhaustion, mental distancing, cognitive impairment, and emotional impairment). According to the cutoffs defined by Schaufeli in Ref. [18], a total core score less than 2.59 signifies the absence of burnout syndrome, a total core score between 2.59 and 3.01 means the predisposition to burnout, and a total core score greater than or equal to 3.02 means the presence of burnout. As can be seen in Table 2, based on the total core score plus SD, each studied group is prone to burnout.

**Table 2.** BAT-23 test results.

| Parameter | Oncology | Cardiology | Intensive Care Unit | Occupational Health | Residents |
|---|---|---|---|---|---|
| Total core score (Mean ± SD) | 2.12 ± 0.48 | 2.16 ± 0.49 | 2.27 ± 0.52 | 2.46 ± 0.55 | 2.12 ± 0.56 |
| Exhaustion (Mean ± SD) | 2.72 ± 0.68 | 2.65 ± 0.60 | 2.75 ± 0.57 | 2.88 ± 0.84 | 2.64 ± 0.69 |
| Mental distancing (Mean ± SD) | 1.81 ± 0.69 | 1.86 ± 0.67 | 2.03 ± 0.61 | 2.28 ± 0.54 | 1.81 ± 0.63 |
| Cognitive impairment (Mean ± SD) | 1.70 ± 0.47 | 1.93 ± 0.59 | 2.00 ± 0.67 | 2.30 ± 0.48 | 1.85 ± 0.54 |
| Emotional impairment (Mean ± SD) | 2.26 ± 0.48 | 2.18 ± 0.59 | 2.31 ± 0.53 | 2.37 ± 0.66 | 2.20 ± 0.69 |

**Table 3.** Test of effects between subjects for BAT subscales.

| Source | Dependent Variable | Type III Sum of Squares | df | Mean Square | F | Sig. | Partial Eta Squared |
|---|---|---|---|---|---|---|---|
| Intercept | Exhaustion | 1099.24 | 1 | 1099.24 | 2396.58 | <0.001 | 0.94 |
| | Mental distancing | 566.81 | 1 | 566.81 | 1422.73 | <0.001 | 0.91 |
| | Cognitive impairment | 565.61 | 1 | 565.61 | 1814.33 | <0.001 | 0.92 |
| | Emotional impairment | 758.72 | 1 | 758.72 | 2119.85 | <0.001 | 0.93 |
| Section | Exhaustion | 0.96 | 4 | 0.24 | 0.53 | 0.72 | 0.01 |
| | Mental distancing | 4.31 | 4 | 1.08 | 2.71 | 0.03 | 0.07 |
| | Cognitive impairment | 5.43 | 4 | 1.36 | 4.36 | 0.002 | 0.11 |
| | Emotional impairment | 0.67 | 4 | 0.17 | 0.47 | 0.76 | 0.01 |

Cutoffs were also defined for each BAT subscale [18]. So, on the exhaustion subscale, a score lower than 3.06 signifies the absence of the burnout syndrome; a score between 3.06 and 3.30 indicates the predisposition to burnout; and a score greater than or equal to 3.31 indicates the presence of burnout. Related to exhaustion, as can be seen from Table 2 and based on the average score plus SD, each studied group reaches the burnout region, with the exception of the cardiology group, which is prone to burnout.

On the mental distancing scale, a score lower than 2.10 signifies the absence of the burnout syndrome; a score between 2.10 and 3.29 indicates the predisposition to burnout; and a score greater than or equal to 3.30 indicates the presence of burnout. Therefore, related to mental distancing and based on the average score plus SD, each studied group reaches the region where it is considered to be prone to burnout.

The cutoffs on the cognitive impairment scale allow inclusion in the three categories, as follows: a score lower than 2.70 signifies the absence of the burnout syndrome; a score between 2.70 and 3.09 indicates the predisposition to burnout; and a score greater than or equal to 3.10 indicates the presence of burnout. Therefore, related to cognitive impairment and taking into account the average score plus SD, only the extent of the occupational health group is observed in the region where it is assumed to be prone to burnout, while for the other groups, the absence of the burnout syndrome is observed.

On the emotional impairment scale, a score lower than 2.30 signifies the absence of burnout syndrome; a score between 2.30 and 3.09 indicates the predisposition to burnout; and a score greater than or equal to 3.10 indicates the presence of burnout. Therefore, related to emotional impairment and based on the average score plus SD, each group is prone to burnout.

A multivariate analysis of variance (MANOVA) was used to simultaneously see the effect of the independent variable *section* (*oncology, cardiology, intensive care unit, occupational health, and the group of residents*) on the dependent variables (BAT subscales—*exhaustion, mental distancing, cognitive impairment,* and *emotional impairment*), and the results of the test are presented in Table 3.

Based on the test results in Table 3, one can conclude that there is no significant effect of the *section* on *exhaustion* ($p = 0.72$) or on *emotional impairment* ($p = 0.76$). There is a significant effect of the *section* on *mental distancing* ($F = 2.71$ and $p = 0.03$) with a medium effect size (partial eta squared = 0.07). There is a significant effect of *the section* on *cognitive impairment* ($F = 4.36$ and $p = 0.002$) with a medium effect size (partial eta squared = 0.11).

The effect of the *section* on *mental distancing* is determined by the occupational health group. The effect of the *section* on *cognitive impairment* is also due to the occupational health group (see Table 2).

The effect on the *mental distancing* and *cognitive impairment* scales is probably related to age and seniority (which are the highest in all study groups, i.e., $50.30 \pm 6.51$ years and $27.39 \pm 9.14$ years, respectively).

### 3.2. Test Results for COPE Questionnaire

Health professionals often resort to numerous coping methods in order to maintain their good mental health for as long as possible. Consciously or not, probably depending on the type of personality, education, work, and life experience, they use different methods of adaptation to stressful or overwhelming situations at work. Coping mechanisms are essential to helping healthcare workers manage the challenges they face daily.

Table 4 shows the results obtained after calculating the answers to the COPE questionnaire (mean and standard deviation for each coping mechanism and for each study group). Thus, it can be observed that in the group of cardiologists, the most frequently used mechanisms are active coping, planning, positive reinterpretation and growth, elimination of competing activities, but also the use of social and emotional instrumental support, and the religious approach. The least used coping mechanisms are those of denial, behavioral passivity, and drug and alcohol abuse.

In the case of the oncology group, the most frequently used, in descending order, are as follows: planning, social emotional support, positive reinterpretation and growth, social instrumental support, but also the religious approach, and the active coping mechanisms. The less-used coping mechanisms are those of behavioral passivity, denial, and drug and alcohol abuse.

In the intensive care unit group, the most frequently used mechanisms are planning, positive reinterpretation and growth, active coping, instrumental social and emotional support, and acceptance. The much less used coping mechanisms are those of denial, behavioral passivity, and drug and alcohol abuse.

In the occupational health group, we observe that the most frequently used mechanisms are planning, active coping, positive reinterpretation and growth, acceptance, and the religious approach. Denial mechanisms, humor, and drug and alcohol abuse are less used.

In the case of the resident group, there are high scores for planning, positive reinterpretation and growth, social instrumental support, active coping, and social emotional support. Denial mechanisms, behavioral passivity, and drug and alcohol abuse are less commonly used.

**Table 4.** COPE questionnaire results.

| Coping Mechanism | Oncology | Cardiology | Intensive Care Unit | Occupational Health | Residents |
|---|---|---|---|---|---|
| Active coping (Mean $\pm$ SD) | $2.82 \pm 0.70$ | $2.94 \pm 0.69$ | $2.93 \pm 0.61$ | $3.05 \pm 0.57$ | $3.06 \pm 0.47$ |
| Planning (Mean $\pm$ SD) | $3.17 \pm 0.59$ | $3.22 \pm 0.57$ | $3.10 \pm 0.57$ | $3.13 \pm 0.55$ | $3.44 \pm 0.44$ |
| Elimination of competing activities (Mean $\pm$ SD) | $2.72 \pm 0.54$ | $2.82 \pm 0.53$ | $2.70 \pm 0.59$ | $2.77 \pm 0.63$ | $2.74 \pm 0.37$ |
| Restrictions to action (Mean $\pm$ SD) | $2.73 \pm 0.48$ | $2.71 \pm 0.54$ | $2.71 \pm 0.42$ | $2.87 \pm 0.34$ | $2.68 \pm 0.46$ |
| Social instrumental support (Mean $\pm$ SD) | $2.93 \pm 0.79$ | $2.89 \pm 0.61$ | $2.84 \pm 0.55$ | $2.82 \pm 0.57$ | $3.11 \pm 0.65$ |
| Social emotional support (Mean $\pm$ SD) | $3.11 \pm 0.63$ | $2.77 \pm 0.71$ | $2.91 \pm 0.59$ | $2.66 \pm 0.62$ | $2.94 \pm 0.82$ |

**Table 4.** *Cont.*

| Coping Mechanism | Oncology | Cardiology | Intensive Care Unit | Occupational Health | Residents |
|---|---|---|---|---|---|
| Positive reinterpretation and growth (Mean ± SD) | 3.03 ± 0.60 | 3.17 ± 0.57 | 3.08 ± 0.55 | 2.99 ± 0.63 | 3.39 ± 0.50 |
| Acceptance (Mean ± SD) | 2.76 ± 0.64 | 2.71 ± 0.64 | 2.77 ± 0.53 | 2.98 ± 0.33 | 2.78 ± 0.56 |
| Denial (Mean ± SD) | 2.07 ± 0.60 | 1.82 ± 0.53 | 2.01 ± 0.57 | 2.04 ± 0.48 | 1.93 ± 0.57 |
| Emotional discharge (Mean ± SD) | 2.30 ± 0.60 | 2.01 ± 0.51 | 2.45 ± 0.56 | 2.53 ± 0.58 | 2.33 ± 0.65 |
| Religious approach (Mean ± SD) | 2.85 ± 0.84 | 2.61 ± 1.18 | 2.59 ± 0.90 | 2.95 ± 0.73 | 2.49 ± 0.93 |
| Mental passivity (Mean ± SD) | 2.55 ± 0.58 | 2.28 ± 0.59 | 2.41 ± 0.62 | 2.38 ± 0.51 | 2.31 ± 0.58 |
| Behavioral passivity (Mean ± SD) | 2.12 ± 0.55 | 1.86 ± 0.64 | 1.91 ± 0.58 | 2.23 ± 0.62 | 1.71 ± 0.52 |
| Humor (Mean ± SD) | 2.23 ± 0.67 | 2.03 ± 0.73 | 2.06 ± 0.78 | 1.65 ± 0.56 | 2.05 ± 0.75 |
| Drug/alcohol intake (Mean ± SD) | 1.36 ± 0.74 | 1.20 ± 0.50 | 1.21 ± 0.65 | 1.13 ± 0.34 | 1.20 ± 0.46 |

In conclusion, we observe that mechanisms based on problem solving (planning, positive reinterpretation, growth, and active coping) are often used in all studied groups, following coping mechanisms focused on emotion (acceptance, religious approach) and coping based on social support (use of instrumental social and emotional support, expression of emotions). In all studied groups, we observe a less frequent use of avoidance coping mechanisms (denial, mental, and behavioral passivity) and the use of drugs and alcohol as being the least used by all study participants. These findings are consistent with other recent studies on healthcare workers [21,22].

A multivariate analysis of variance (MANOVA) was used to simultaneously see the effect of the independent variable *section* (*oncology, cardiology, intensive care unit, occupational health, and the group of residents*) on the dependent variables (COPE questionnaire scales), and the results of the test are presented in Table 5.

Based on the results of the test in Table 5, one can conclude that there is a significant effect of the *section* on *positive reinterpretation and growth* (F = 2.77 and $p = 0.03$) with a medium effect size (partial eta squared = 0.07), the highest score being recorded in the case of the resident group. There is a significant effect of the *section* on *emotional discharge* (F = 2.88 and $p = 0.02$) with a medium effect size (partial eta squared = 0.07), the lowest score being recorded in the case of the group of cardiologists. There is a significant effect of the *section* on *behavioral passivity* (F = 4.08 and $p = 0.004$) with a medium effect size (partial eta squared = 0.09), the highest score being recorded in the case of the occupational health group. There is no significant effect of the *section* on the other COPE scales ($p > 0.05$).

**Table 5.** Test of effects between subjects for COPE scales.

| Source | Dependent Variable | Type III Sum of Squares | df | Mean Square | F | Sig. | Partial Eta Squared |
|---|---|---|---|---|---|---|---|
| Intercept | Active coping | 1295.55 | 1 | 1296 | 3516 | <0.001 | 0.96 |
| | Planning | 1525.28 | 1 | 1525 | 5235 | <0.001 | 0.97 |
| | Elimination of competitive activity | 1118.50 | 1 | 1119 | 4047 | <0.001 | 0.96 |
| | Restraint from action | 1110.28 | 1 | 1110 | 5319 | <0.001 | 0.97 |
| | Social support | 1257.95 | 1 | 1258 | 3009 | <0.001 | 0.95 |
| | Emotional support | 1224.99 | 1 | 1225 | 2585 | <0.001 | 0.95 |
| | Positive reinterpretation and growth | 1449.02 | 1 | 1449 | 4577 | <0.001 | 0.96 |
| | Acceptance | 1158.12 | 1 | 1158 | 3728 | <0.001 | 0.96 |
| | Denial | 576.20 | 1 | 576.20 | 1832 | <0.001 | 0.93 |
| | Emotional discharge | 798.05 | 1 | 798.05 | 2307 | <0.001 | 0.94 |
| | Religious approach | 1076.07 | 1 | 1076 | 1258 | <0.001 | 0.89 |
| | Mental passivity | 841.38 | 1 | 841.38 | 2487 | <0.001 | 0.94 |
| | Behavioral passivity | 570.88 | 1 | 570.88 | 1737 | <0.001 | 0.92 |
| | Drug/alcohol abuse | 220.51 | 1 | 220.51 | 682.24 | <0.001 | 0.82 |
| | Humor | 594.71 | 1 | 594.71 | 1169 | <0.001 | 0.88 |
| Section | Active coping | 1.28 | 4 | 0.32 | 0.86 | 0.48 | 0.02 |
| | Planning | 2.76 | 4 | 0.69 | 2.37 | 0.05 | 0.06 |
| | Elimination of competitive activity | 0.25 | 4 | 0.06 | 0.23 | 0.92 | 0.006 |
| | Restraint from action | 0.59 | 4 | 0.15 | 0.71 | 0.58 | 0.02 |
| | Social support | 1.85 | 4 | 0.46 | 1.10 | 0.35 | 0.03 |
| | Emotional support | 3.19 | 4 | 0.79 | 1.68 | 0.15 | 0.04 |
| | Positive reinterpretation and growth | 3.52 | 4 | 0.88 | 2.77 | 0.03 | 0.07 |
| | Acceptance | 1.03 | 4 | 0.26 | 0.82 | 0.51 | 0.02 |
| | Denial | 1.11 | 4 | 0.28 | 0.88 | 0.47 | 0.02 |
| | Emotional discharge | 3.99 | 4 | 0.99 | 2.88 | 0.02 | 0.07 |
| | Religious approach | 4.33 | 4 | 1.08 | 1.27 | 0.28 | 0.03 |
| | Mental passivity | 1.35 | 4 | 0.34 | 1.00 | 0.41 | 0.02 |
| | Behavioral passivity | 5.37 | 4 | 1.34 | 4.08 | 0.004 | 0.09 |
| | Drug/alcohol abuse | 0.86 | 4 | 0.22 | 0.68 | 0.60 | 0.02 |
| | Humor | 4.72 | 4 | 1.18 | 2.32 | 0.06 | 0.06 |

*3.3. Pulse Wave Velocity (PWV) Results*

The mean and standard deviation (SD) for the PWV values in the studied groups are represented in Table 6.

**Table 6.** Pulse wave velocity results of the study groups.

| Parameter | Oncology | Cardiology | Intensive Care Unit | Occupational Health | Residents |
|---|---|---|---|---|---|
| PWV (Mean ± SD) (m/s) | 7.77 ± 1.31 | 9.66 ± 2.10 | 8.72 ± 2.14 | 9.69 ± 1.45 | 7.46 ± 1.30 |
| Normal PWV (Mean ± SD) (m/s) Ref. [23] | 6.5 ± 1.85 | 7.2 ± 1.3 | 6.5 ± 1.85 | 8.3 ± 1.9 | 6.2 ± 0.7 |

The reference values for arterial stiffness in the case of the healthy European population and in the presence of cardiovascular risk factors are presented in Ref. [23], per age decade. Table 6 also shows the normal values of PWV, taking into account the age decades in which subjects in each study group fall, according to reference [23]. One can observe that all study groups have slightly increased PWV values compared to normal ones.

A multivariate analysis of variance was used to determine the effect of the independent variable *section* (*oncology, cardiology, intensive care unit, occupational health, and the group of residents*) on PWV, and the results of the test are presented in Table 7. Based on the results of the test from Table 7, one can conclude that there is a significant effect of the *Section* on PWV (F = 11.36 and $p < 0.05$) with a large effect size (partial eta squared = 0.23). The effect of the *section* on PWV is due to the occupational health and cardiology groups.

**Table 7.** Tests of effects between subjects—dependent variable PWV.

| Source | Type III Sum of Squares | df | Mean Square | F | Sig. | Partial Eta Squared |
|---|---|---|---|---|---|---|
| Corrected model | 127.96 [a] | 4 | 31.99 | 11.36 | <0.001 | 0.23 |
| Intercept | 11,083.53 | 1 | 11,083.53 | 3935.63 | <0.001 | 0.96 |
| Section | 127.96 | 4 | 31.99 | 11.36 | <0.001 | 0.23 |
| Error | 419.61 | 149 | 2.82 | | | |
| Total | 11,635.01 | 154 | | | | |
| Corrected total | 547.58 | 153 | | | | |

[a] R squared = 0.234 (adjusted R squared = 0.213).

### 3.4. BAT-PWV Correlations

The PWV analysis is a tool for assessing the cardio-vascular risk, and the relationship between burnout and PWV can be analyzed through the lens of the link between stress and the risk of cardiovascular pathology as a result of stress. Table 8 shows the Pearson's correlation coefficients (*r*) and the significance (*p*) for the correlations between PWV, age, body mass index (BMI), and BAT facets, for each study group.

From Table 8, one can observe that in the total group (which consists of total subjects from all groups), there are some positive correlations (but weak, i.e., $r < 0.3$) between PWV and some of the BAT subscales, e.g., with the total core score ($r = 0.191$), mental distancing ($r = 0.177$), and cognitive impairment ($r = 0.290$). In the oncology group, there is a positive, moderate correlation between PWV and emotional impairment ($r = 0.353$).

In Table 8, one can also observe that in the total group, there is a good positive correlation between PWV and age ($r = 0.552$) as well as a moderate correlation between PWV and BMI ($r = 0.362$).

**Table 8.** Correlations between PWV, age, BMI, and BAT facets.

| PWV | | Age | Total Core Score | Exhaustion | Mental Distancing | Cognitive Impairment | Emotional Impairment | BMI |
|---|---|---|---|---|---|---|---|---|
| Total group (total subjects, in all groups) | *r* | 0.552 | 0.191 | 0.065 | 0.177 | 0.290 | 0.132 | 0.362 |
| | *p* | <0.001 | 0.018 | 0.424 | 0.028 | <0.001 | 0.102 | <0.001 |
| Intensive care unit | *r* | 0.607 | 0.276 | 0.273 | 0.262 | 0.232 | 0.194 | 0.309 |
| | *p* | <0.001 | 0.120 | 0.124 | 0.140 | 0.194 | 0.278 | 0.080 |
| Cardiology | *r* | 0.473 | −0.075 | −0.252 | −0.134 | 0.220 | −0.066 | 0.294 |
| | *p* | 0.017 | 0.720 | 0.225 | 0.523 | 0.291 | 0.754 | 0.152 |
| Oncology | *r* | 0.258 | 0.314 | 0.270 | 0.126 | 0.340 | 0.353 | 0.229 |
| | *p* | 0.146 | 0.076 | 0.129 | 0.485 | 0.053 | 0.044 | 0.198 |
| Occupational medicine | *r* | 0.276 | 0.192 | 0.062 | 0.170 | 0.184 | 0.292 | 0.275 |
| | *p* | 0.203 | 0.380 | 0.777 | 0.438 | 0.401 | 0.176 | 0.204 |
| Residents | *r* | 0.049 | 0.011 | −0.158 | 0.118 | 0.090 | 0.018 | 0.175 |
| | *p* | 0.765 | 0.945 | 0.332 | 0.467 | 0.579 | 0.914 | 0.278 |

*r*—Pearson's correlation coefficient; *p*—significance.

## 4. Discussion

The purpose of this study was to evaluate the presence of burnout syndrome, find out the most commonly used coping methods, and assess the possible relationship between PWV and the work place of healthcare workers in some university clinic sections. PWV is a predictive parameter for cardiovascular diseases [13,14] and its evaluation in correlation with work conditions gives us information about possible cardiovascular diseases, possibly induced by conditions at the workplace.

The results of the BAT-23 burnout evaluation, which take into account the average value and the standard deviation for the *total core score* (Mean ± SD, in Table 2), show that the study groups have subjects prone to burnout. Similar results with small numbers of subjects at burnout were reported in Ref. [24], using the BAT-23, among healthcare workers in Brazil, but they worked in the infirmary and intensive care sectors that cared for patients diagnosed with COVID-19, as well as those for other diseases.

In another study [25], carried out with BAT-23 in Korean nurses working in COVID-19 departments and other departments, it is shown that according to the *total core score*, 45.5% of subjects are burnout. The main symptom in this study was *exhaustion* (74.5% of the subjects).

As reported in Ref. [24], *exhaustion* and *emotional impairment* were the main symptoms presented. In our study, each studied group reaches the burnout region, with the exception of the cardiology group, which is prone to burnout, with respect to *exhaustion.* The result in the cardiology group may be due to organizational rules or a subjective perception.

Regarding *emotional impairment*, in our study groups, based on the average score plus SD, each group is prone to burnout. Unlike the results in Ref. [24], in our study, larger scores for *mental distancing* were recorded in the intensive care unit group and in the occupational health group. If in the case of the subjects who work in the intensive care unit, the *mental distancing* can be attributed to the severity of the cases, in the group of subjects who work in occupational health, the *mental distancing* can be explained by the older average age (50.30 ± 6.51 years). However, as Schaufeli and De Witte mentioned [26], mental distancing serves as an ineffective coping strategy to reduce exhaustion.

In a study published in Ref. [27], performed with the *Maslach Burnout Inventory-General Survey*, on healthcare workers during the COVID-19 pandemic in Italy, the risk of burnout was higher in professionals with shorter seniority at the workplace. In our study, conducted after the COVID-19 pandemic with BAT-23, the results show that the group of residents with a work experience of 3.05 ± 2.59 years has good BAT scores with respect to subjects from other groups with greater work experience. In our study, the highest BAT scores are recorded for subjects in the occupational health group (see Table 2), the group with

the highest work experience (27.39 ± 9.13 years) and the highest age (50.30 ± 6.51 years). Results similar to ours are reported in ref. [28], in which BAT-12 was used, and it found subjects with a higher seniority to have a higher burnout score.

The analysis of the BAT-23 scores according to the *section* (workplace) shows that there is no significant effect of the *section* on *exhaustion* and *emotional impairment*. There is a significant effect of the *section* on *mental distancing* (which is due to the subjects in the occupational health group). Also, there is a significant effect of the *section* on *cognitive impairment* (which is due to subjects in the occupational health group). Thus, according to the analysis presented in Table 2, it turns out that perhaps only subjects from the intensive care unit and occupational health groups are affected by the workplace.

Healthcare workers employ various coping strategies to manage stress and maintain their mental health during their daily activities. Coping strategies can be categorized into two types: individual-level coping and organizational-level coping. In our studied groups, individual coping was the only type used. According to a study conducted by Chen et al. [29], healthcare workers who used individual-level coping had better mental health outcomes than those who did not.

In our study group, planning, positive reinterpretation, growth, and active coping have the highest mean score, being the coping strategies most frequently used by all participants. Similar results were reported by other authors [30–32].

Effective planning is a cornerstone of positive coping through active engagement, which involves the development of strategies to systematically address and manage stressors. Planning allows people to break down overwhelming challenges into manageable tasks, providing a sense of control and purpose. This proactive approach helps people anticipate potential obstacles, consider alternative solutions, and make informed decisions. According to Lazarus and Folkman's transactional model of stress and coping [33], problem-focused coping, which includes planning, is particularly effective in situations where the stressor can be changed or eliminated.

Positive reinterpretation involves reframing the meaning of a stressful situation in a more positive or constructive light. This cognitive coping strategy is rooted in the idea that one's perspective significantly influences emotional reactions. By consciously altering the interpretation of a stressor, individuals can reduce the emotional impact and foster a more adaptive response [34]. Positive reinterpretation is aligned with the cognitive-behavioral approach, emphasizing the interplay between thoughts, emotions, and behaviors.

Embracing growth as a coping strategy involves finding meaning and personal development in the face of adversity. Post-traumatic growth, a concept introduced by Tedeschi and Calhoun [35], suggests that individuals can experience positive psychological changes following highly challenging experiences. This growth may manifest as increased resilience, a deeper appreciation for life, and a greater sense of personal strength. Actively seeking opportunities for personal development in the aftermath of stressors can contribute to long-term well-being.

Other coping mechanisms used by the subjects of our study groups focused on emotion (acceptance and religious approach) and coping based on social support (use of instrumental social and emotional support and expression of emotions).

Acceptance as a coping strategy that involves acknowledging and embracing the reality of a situation without attempting to change it. Rooted in mindfulness and the principles of acceptance and commitment therapy, this approach encourages individuals to let go of futile attempts to control uncontrollable circumstances. By accepting the inherent uncertainties of life, people can reduce the emotional burden of stressors and focus on adapting to the situation at hand [36].

Religious beliefs and practices have long been recognized as potent resources for coping with life's challenges. Many religions provide frameworks that offer solace, meaning, and a sense of purpose in the face of adversity. Drawing on faith, prayer, and community support, religious coping can help people find meaning in suffering and cultivate resilience.

Research suggests that religious coping is associated with positive mental health outcomes and can be a valuable asset in times of distress [37].

Emotional support, whether from friends, family, or a broader community, is a crucial element in coping with stress. Social support provides a buffer against the negative effects of life events and fosters a sense of connection. Emotional support encompasses the provision of empathy, understanding, and encouragement, creating a safe space for people to express their feelings. Studies consistently highlight the positive impact of social support on mental health and well-being [38].

Avoidance coping with mental passivity has unexpectedly high average scores (between 2.31 and 2.71) in our study groups. This form of coping may involve withdrawing from decision-making, ignoring problems, or engaging in escapist behaviors. Mental passivity can contribute to a cycle of inaction and exacerbate feelings of powerlessness [33]. However, taking into account that planning, positive reinterpretation, growth, and active coping have the highest mean score in our study group, we may conclude that a relatively large score of mental passivity can be explained by fatigue, close to exhaustion. To support this statement, we note that behavioral passivity has low average values.

The analysis of COPE results relative to *section* (workplace) shows that there is a significant effect of the section on behavioral passivity (F = 4.08 and *p* = 0.004) with a medium effect size (partial eta squared = 0.09). This is due to the subjects in the occupational health group and can be explained by their older average age (50.30 ± 6.51 years).

Although mental passivity and behavioral passivity may offer a temporary respite from the discomfort of facing challenging situations, their long-term consequences for mental health can be detrimental. Persistent use of mental passivity and behavioral passivity has been linked to increased stress, anxiety, and depression as individuals miss opportunities for growth and fail to address underlying issues [39].

Drug and alcohol abuse was the COPE dimension with the lowest mean score, meaning that it was the coping strategy least reported by all participants. As mentioned by other authors [40,41], this result should be viewed with caution because many people do not recognize this coping strategy due to the possible association with their social stigmatization.

Occupational stress and burnout have long been assumed to be significant contributors to a range of physical and mental health issues among employees. One emerging tool of research on the physiological consequences of chronic stress is the study of pulse wave velocity (PWV). PWV is considered an indicator of arterial stiffness, with higher PWV values indicating increased stiffness and potential cardiovascular risks [14,42]. Apart from various cardiovascular diseases, arterial stiffness and high values of PWV are associated with aging (the higher the age, the higher the PWV value) [23,43].

The relationship between PWV and burnout can be understood through the stress-response pathway. Chronic exposure to stress, such as that experienced in the workplace, activates the sympathetic nervous system and the hypothalamic–pituitary–adrenal (HPA) axis, leading to increased levels of stress hormones like cortisol and adrenaline [44]. These hormones, in turn, contribute to arterial stiffness and endothelial dysfunction, key factors that influence PWV [45]. Empirical studies have provided insights into the connection between occupational stress, burnout, and pulse wave velocity. For example, a study by Kamarck et al. [46,47] found that higher levels of chronic work-related stress were associated with increased PWV. Similarly, Tsai et al. [48] demonstrated that burnout was independently associated with higher PWV, suggesting a potential link between psychological well-being and cardiovascular health.

In our study groups, one can observe slightly increased PWV values compared to normal ones. There are positive correlations between PWV and only some components of BAT (weak correlation with *mental distancing* and with *cognitive impairment*). Only in the oncology group is there a positive, moderate correlation between PWV and *emotional impairment.* Thus, according to our study, there is no clear connection between burnout and PWV. This result can be explained by the fact that burnout is rather a so-called momentary state (it may occur after a stressful period of the order of months), whilst the PWV value

(respectively arterial stiffness) is the result of chronic factors (to which the subject is exposed over a long period of time, of the order of years). In support of this statement, we find from our study that PWV correlates better with age and BMI than with the BAT subscales.

## 5. Limitations and Future Directions

### 5.1. Study Limitations

This study, in which BAT-23 was used to evaluate burnout syndrome and cutoffs, is the first conducted in Romania on medical personnel; however, there are some limitations and future directions to follow. The number of participants in each batch was relatively small, and the majority of subjects were females. Apart from demographic data, the workplace, and measurements made with the arteriograph (objective data), the rest of the data are subjective because we used self-assessment questionnaires for the state of burnout and coping mechanisms. It is possible that some answers are exaggerated and others diminished, influenced by certain situations or actual states. However, our results are similar to those of other studies, but a unitary application is needed on a much larger population of the same new burnout evaluation questionnaire (BAT-23) [2].

### 5.2. Future Research

It is necessary to further research the relationship between the workplace and burnout and its subscales, the effectiveness of coping mechanisms to preserve the mental health of health workers, the effectiveness of the medical act, and the sustainability of the medical career.

Understanding the relationship between the well-being of healthcare workers and the quality of patient care is crucial. Future research can explore how burnout and coping mechanisms influence clinical decision-making, patient satisfaction, and overall healthcare outcomes.

This study should continue with the concrete application and assessment of the effectiveness of measures or therapies for healthcare workers who are found to have burnout, such as counseling, psychological help, recommendation of physical exercises, training in coping strategies, and mindfulness. Investigating the effectiveness of various interventions designed to alleviate burnout and enhance coping strategies is crucial. This may involve testing the impact of mindfulness programs, resilience training, support groups, or organizational changes to determine which interventions are most beneficial for healthcare professionals.

Investigating the role of education and training programs in the prevention of burnout and the promotion of effective coping is another approach to improving the professional training curriculum in the field of healthcare in different career stages.

Further studies should focus on investigating the benefits of applying the BAT-23 instrument to assess burnout. As we see, the development and consequences of each burnout dimension should be examined. Investigating individual differences in responses to burnout and coping interventions can help tailor strategies to specific personality traits, job roles, or personal circumstances. Personalized approaches may produce more effective outcomes for healthcare workers.

### 5.3. Practical Recommendations

When addressing burnout among medical personnel, it is important to consider both organizational- and individual-level interventions. Here are some medical measures and strategies to help prevent and alleviate burnout among healthcare professionals:

- Regular Health Check-ups—providing access to preventive care and screenings.
- Mental Health Support—provide confidential counseling services to address the emotional and psychological challenges that medical personnel may face.
- Education on Stress Management—by training sessions and workshops on stress management techniques, coping strategies, and mindfulness practices.

- Fatigue Management—implementing policies to manage fatigue, including guidelines on work hours, breaks, and rest periods.
- Recognition and Appreciation—acknowledgement and appreciation of the efforts of medical personnel through regular feedback and recognition programs.
- Conflict Resolution Support—providing training on communication and conflict resolution skills.
- Accessible leadership - foster a sense of trust and collaboration between leadership and medical personnel.

## 6. Conclusions

The present study provides some insights into how the workplace can influence the level of burnout, coping strategies, and possible correlations with pulse wave velocity.

According to our study, it seems that many healthcare workers are prone to burnout or reach burnout.

In relation to exhaustion, each studied group reached the burnout region, with the exception of the cardiology group, which was prone to burnout. Exhaustion of healthcare workers can have significant consequences, both for individuals experiencing burnout and for the healthcare system as a whole. Fatigue and burnout can increase the risk of medical errors. Mistakes in diagnosis, medication administration, and other critical tasks can have serious consequences for patient safety and outcomes. Chronic exhaustion and burnout can take a toll on mental and physical health. This can manifest as symptoms such as anxiety, depression, insomnia, and other stress-related conditions, further exacerbating the overall health burden on individuals.

On the mental distancing subscale, each studied group is prone to burnout. Mental distancing, also known as depersonalization, can negatively impact the quality of patient care, leading to decreased patient satisfaction as healthcare professionals may become less attuned to the individual needs and concerns of those they are treating. Mental distancing can extend beyond the patient–provider relationship and affect interactions with colleagues and other healthcare staff. Healthcare workers who are emotionally detached may struggle to collaborate effectively, leading to strained interpersonal relationships within the healthcare team. This can hinder communication, teamwork, and overall cohesion in providing patient care.

Cognitive impairment can lead to difficulty concentrating, memory loss, and impaired decision-making. These cognitive challenges significantly increase the risk of medical errors, such as misdiagnosis, medication errors, or incorrect treatment plans. The consequences of medical errors can be severe and negatively affect patient safety and outcomes. Our study shows that the study groups are not affected by burnout in terms of cognitive impairment, with the exception of the occupational health group, which is prone to burnout. Taking into account that the occupational health group has the highest average age, we consider that the age factor is the one that induced higher values of the level of burnout on the cognitive impairment subscale. In this respect, avoiding long shifts or night shifts may be beneficial for elder healthcare workers.

Related to emotional impairment, each group is prone to burnout. Emotional impairment can negatively impact the patient–provider relationship, potentially leading to lower patient satisfaction and compromised communication. Also, difficulty managing one's emotions may result in increased workplace stress and a less supportive work environment. Healthcare workers experiencing emotional impairment can withdraw socially and isolate themselves from colleagues and support networks. Professional isolation can also impact the exchange of knowledge and hinder the sharing of experiences, which are important components of ongoing professional development and support within the healthcare community.

While there are potential solutions to burnout, more research is needed to determine their effectiveness. Healthcare organizations must prioritize addressing burnout to promote

the well-being of healthcare workers, improve the quality of care for patients, and ensure the sustainability of the medical career.

Various coping strategies are frequently used in the studied groups, such as positive reinterpretation and growth, planning, and active coping, but they are not influenced by the workplace (department—*section*).

As organizations and healthcare professionals strive to address the increasing prevalence of burnout, understanding the physiological markers (such as PWV) becomes imperative. The PWV values of the medical workers in our study groups are higher than the European average for the same decades of age. This can be a problem related to the profession itself and not necessarily related to the workplace (section/department) of caregivers. Additionally, according to our study, there is a weak or moderate correlation between PWV and some components of BAT-23 related to burnout, with PWV being better correlated with age and BMI.

**Author Contributions:** Conceptualization, I.M. and C.F.-M.; methodology, I.M.; software, V.S. and C.N.M.; validation, S.U. and C.F.-M.; formal analysis, I.M., data curation, C.N.M.; investigation, I.M.; writing—original draft preparation, I.M and C.N.M.; writing—review and editing, S.U.; visualization, C.F.-M.; supervision, S.U. All authors have read and agreed to the published version of the manuscript.

**Funding:** This research received no external funding.

**Institutional Review Board Statement:** The study was conducted according to the guidelines of the Declaration of Helsinki and approved by the Ethics Committee of the "Victor Babes" University of Medicine and Pharmacy, Timisoara, No. 59/22.12.2021.

**Informed Consent Statement:** Informed consent was obtained from all subjects involved in the study. Written informed consent has been obtained from the patients to publish this paper.

**Data Availability Statement:** The study protocol and supplementary information will be provided upon request.

**Acknowledgments:** We thank all the subjects who kindly participated in the study.

**Conflicts of Interest:** The authors declare no conflicts of interest.

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
