# Peer review of "An Analysis of Burnout, Coping, and Pulse Wave Velocity in Relation to the Workplace of Healthcare Workers for the Sustainability of the Medical Career"

_sustainability, doi:10.3390/su16030997_

Round 1

Reviewer 1 Report

Comments and Suggestions for Authors

The thesis that there is currently an increasing interest in the sustainability of a medical career, which is why it is important to assess professional burnout and analyze coping styles in relation to the workplace and physical and mental symptoms, is fully justified - because professional burnout includes physical, emotional and mental exhaustion, which may negatively impact the quality of a doctor's work and his ability to provide effective help to patients. Furthermore, focusing on the analysis of coping styles in relation to the workplace allows us to identify strategies that help doctors maintain work-life balance...

Reviewer 2 Report

Comments and Suggestions for Authors

There is an increased interest in the sustainability of medical careers, therefore it is important to evaluate burnout and analyze coping styles in relation to the workplace and physical and mental symptoms. This study explores the level of burnout, coping strategies and pulse wave velocity (PWV) using a sample of 154 healthcare workers aged 25-64 years, 83.76% female, divided into five groups (oncology, cardiology, intensive care unit, occupational health and residents). The BAT-23 questionnaire was used for the burnout level and the COPE questionnaire for the coping strategies. PWV was measured with TensioMed arteriograph. Regarding BAT subscales, all studied groups are at burnout for Exhaustion, prone to burnout for Mental distancing, Emotional impairment, and no burnout for Cognitive impairment. On the 4 subscales of the BAT, the MANOVA shows a medium size effect of workplace on mental distancing and on cognitive impairment. All subjects used positive coping strategies. Problem-solving mechanisms are often used, followed by emotion-focused and social support-based coping. PWV has a small positive correlation with mental distancing, cognitive impairment, and emotional impairment. PWV correlates better with age and BMI.  PWV depends on age and BMI rather than burnout. The authors report that high burnout levels are related to the healthcare profession, not to the workplace.  Specific comments follow:

1. You began the abstract with “There is an increased interest in the sustainability of the medical career” which concerned me as to the use of English as there are multiple careers and career paths in the medical arena and the Materials & Methods section began “The participants were enroled voluntary on the basis of informed consent after the purpose of the study was presented and explained face to face to each of them” which added to my concerns about the use of English but fortunately these are exceptions rather than the norm throughout your paper so you should edit these statements however they are not a major problem.

2. I have reviewed many papers which examine burnout and stress in heath care and believe that this paper does provide a unique spin that added to the literature – however you need to integrate it in to the literature of Sustainability [this journal] as well as build the paper further in to the existing broad literature on health care coping, stress, and burnout.

3. I wish that you had recruited a few more health care workers as a sample of 154 not only is unlikely to have a generally acceptable level of power [doing a power analysis in my head you would want a a sample of at least 168 and for stability of results 200 would likely be the minimum needed].  If possible it might be a good idea for you to go to another research site and seek to have at least 200 responses – although this is not a hard requirement but rather a suggestion.

4. When it comes to your analyses I wondered why you chose to do MANOVAs as opposed to  MANCOVAs.

5. I believe that you should add additional specific suggestions for future research.  You have very strong results and intriguing ideas however you likely underestimate the potential importance of your findings and should be able to influence future research in a meaningful way.

Thank you very much for submitting your paper to Sustainability.  I hope that you find these comments and suggestions useful at improving the potential value added to the literature by your paper and that you continue to consider Sustainability as an outlet for your finest works in to the future.

Comments on the Quality of English Language

There are a few examples that I provided in the review where the paper should be edited for English usage.

Reviewer 3 Report

Comments and Suggestions for Authors

The paper entitled “An Analysis of Burnout, Coping, and PWV in Relation to the Workplace of Healthcare Workers for the Sustainability of the Medical Career” explores the level of burnout, coping strategies and pulse wave velocity. The collected data included 154 healthcare workers aged 25-64 years, and was divided into five groups (oncology, cardiology, intensive care unit, occupational health and residents).

Based on the performed review of the paper the following comments can be provided:

Abstract. The importance of the topic of this research, data collection approaches and research methods are identified, but the main practical and theoretical contributions of this study are not identified. Please revise the abstract.

Besides that, explain the possibilities of implementing the obtained results in the theory, practically by organizations/hospitals and by the state during policy formation.

“The BAT- questionnaire was used for the burnout level and the COPE questionnaire for the coping strategies”

Revise the usage of abbreviations in the abstract. It is suggested that: The first time you use an abbreviation, it's important to spell out the full term and put the abbreviation in parentheses. Then, you can use just the abbreviation in subsequent references after that.

References.

https://www.who.int/news/item/28-05-2019-burn-out-an-occupational-phenomenon-international-classification-of-diseases 498 (accessed on 3 August 2023)

https://www.who.int/news-room/fact-sheets/detail/cardiovascular-diseases-(cvds) (accessed on 11 July 2023)  

Revise the format of references.
